# Enhancing Resistance to Cercospora Leaf Spot in Mung Bean *(Vigna radiata* L.) through *Bradyrhizobium* sp. DOA9 Priming: Molecular Insights and Bio-Priming Potential

**DOI:** 10.3390/plants13172495

**Published:** 2024-09-05

**Authors:** Apisit Songsaeng, Pakpoom Boonchuen, Phongkeat Nareephot, Pongdet Piromyou, Jenjira Wongdee, Teerana Greetatorn, Sukanya Inthaisong, Piyada Alisha Tantasawat, Kamonluck Teamtisong, Panlada Tittabutr, Shusei Sato, Nantakorn Boonkerd, Pongpan Songwattana, Neung Teaumroong

**Affiliations:** 1School of Biotechnology, Institute of Agricultural Technology, Suranaree University of Technology, Nakhon Ratchasima 30000, Thailand; apisit161239@gmail.com (A.S.);; 2Institute of Research and Development, Suranaree University of Technology, Nakhon Ratchasima 30000, Thailand; 3School of Crop Production Technology, Institute of Agricultural Technology, Suranaree University of Technology, Nakhon Ratchasima 30000, Thailandpiyada@sut.ac.th (P.A.T.); 4The Center for Scientific and Technological Equipment, Suranaree University of Technology, Nakhon Ratchasima 30000, Thailand; 5Graduate School of Life Sciences, Tohoku University, Sendai 980-8577, Japan

**Keywords:** mung bean, *Cercospora canescens*, Cercospora leaf spot, *Bradyrhizobium*, bio-priming

## Abstract

Mung bean (*Vigna radiata* L.), a vital legume in Asia with significant nutritional benefits, is highly susceptible to Cercospora leaf spot (CLS) caused by *Cercospora canescens*, leading to significant yield losses. As an alternative to chemical fungicides, bio-priming with rhizobacteria can enhance plant resistance. This study explores the potential of *Bradyrhizobium* sp. strain DOA9 to augment resistance in mung bean against CLS via root priming. The results reveal that short (3 days) and double (17 and 3 days) priming with DOA9 before fungal infection considerably reduces lesion size on infected leaves by activating defense-related genes, including *Pti1*, *Pti6*, *EDS1*, *NDR1*, *PR-1*, *PR-2*, *Prx*, and *CHS*, or by suppressing the inhibition of *PR-5* and enhancing peroxidase (POD) activity in leaves. Interestingly, the Type 3 secretion system (T3SS) of DOA9 may play a role in establishing resistance in *V. radiata* CN72. These findings suggest that DOA9 primes *V. radiata* CN72′s defense mechanisms, offering an effective bio-priming strategy to alleviate CLS. Hence, our insights propose the potential use of DOA9 as a bio-priming agent to manage CLS in *V. radiata* CN72, providing a sustainable alternative to chemical fungicide applications.

## 1. Introduction

Mung bean (*Vigna radiata* L.) is a crucial legume crop widely consumed worldwide. Originating in India, it is predominantly cultivated in Southern and Eastern Asia. Still, it is also grown in some parts of Europe, as well as the warmer regions of Canada and the United States [1,2,3]. With its rich nutritional profile and adaptability to varied climatic conditions, the mung bean has significantly emerged as a valuable food source and a key element of sustainable agricultural systems globally [4,5]. Mung bean also bolsters soil fertility through its capability to fix atmospheric nitrogen in symbiotic association with rhizobial bacteria [6,7]. However, Cercospora leaf spot (CLS) disease, caused by the fungal pathogen *Cercospora canescens*, is a major biotic stress and a critical problem for mung bean production worldwide, causing production losses of up to 50–70% [8,9,10]. The majority of economically vital mung bean cultivars are highly susceptible to this disease. Conventional CLS management mainly relies on chemical fungicides, which are effective, but they impose environmental and health hazards [11,12]. Therefore, there is a pressing need to explore alternative, eco-friendly strategies for controlling CLS in mung beans.

Bio-priming is an alternative strategy for treating plants or seedlings with beneficial microorganisms, such as plant growth-promoting rhizobacteria (PGPR) or fungi, in order to enhance plant growth, stress tolerance, and disease resistance [13,14]. In general, priming with living organisms stimulates the plant immune system through the sensing of secreted molecules or signaling compounds derived from the microorganisms during their interaction. The plant immune system can be divided into two main responses: pattern-triggered immunity (PTI) and effector-triggered immunity (ETI). PTI is activated by pathogen-associated molecular patterns (PAMPs) that recognize the conserved molecular structures of pathogens, such as fungal chitin, *β*-glucan, bacterial flagellin, lipopolysaccharide (LPS), and exopolysaccharides (EPS) to initiate a basal plant defense response [15,16,17,18]. On the other hand, ETI is triggered when plants recognize specific pathogen effectors secreted from the secretion system, particularly via Type 3 secretion system (T3SS) processes by bacterial pathogens. The specific recognition between T3SS effectors through ETI often leads to a localized programmed cell death response known as the hypersensitive response (HR), which restricts pathogen spread by sacrificing infected cells and creating a physical barrier around the infection site [16,19].

Local induction of both PTI and ETI triggers broad-spectrum immunity to subsequent pathogen attacks in distal tissues, a phenomenon called systemic acquired resistance (SAR) [20,21,22]. The key signaling molecule of SAR is salicylic acid (SA), which induces various mechanisms such as lignification to establish structural barriers and the induction of pathogenesis-related gene expression such as *PR-1* (chitinases and glucanases), *PR-5* (thaumatin-like proteins), defensins, and thionins [23,24]. Moreover, the transcriptional coactivators *NPR1* and *EDS1* also play a crucial role in the control of SAR in *Arabidopsis* [25]. On the other hand, induced systemic resistance (ISR) is activated by beneficial microorganisms, such as PGPR and mycorrhizal fungi, that colonize the roots or rhizosphere. ISR is activated by jasmonic acid (JA) and ethylene (ET), the chemical signals for inducing the expression of defense-related genes [26,27,28]. However, the pretreatment of some PGPR on plants, such as *Bacillus cereus*, can activate the ISR response by inducing the expression of defense-related genes associated with both SA-dependent and SA-independent signaling pathways [28,29]. This dual activation of defense mechanisms contributes to the broad-spectrum resistance conferred by ISR against a wide range of pathogens.

In Rhizobium–legume symbiosis, the plant’s defensive responses are crucial in controlling the symbiotic efficiency and managing interaction. Certain molecules involved in this symbiosis can trigger the plant’s immune system, leading to either positive or negative impacts on the initiation and continuation of the symbiotic relationship. One example is EPS, which play a part in the initial recognition and regulation of rhizobia during the early stages of symbiosis [30,31,32]. Rhizobia also feature a T3SS that enables the secretion of effector proteins into the host plant cells. This secretion modulates the plant’s immune responses, either suppressing or activating particular defense pathways to foster the symbiotic interaction [33,34]. Notably, the *Bradyrhizobium* sp. strain DOA9 can form necrotic nodules in the *V. radiata* root, whereas a DOA9 strain lacking T3SS successfully cultivates normal symbiotic nodules [35].

Moreover, T3SS-DOA9 mitigates early defense mechanisms during nodulation in *V. radiata* CN72 roots [36]. In light of these observations, it appears that DOA9 plays a part in triggering plant defense responses in *V. radiata*. This study aims to examine *Bradyrhizobium* sp. DOA9′s potential to heighten resistance traits against CLS disease in mung bean leaves. It reveals that priming roots with *Bradyrhizobium* sp. strain DOA9 is a viable strategy for enhancing mung bean resistance against CLS. Additionally, this study investigates the defense mechanisms activated by DOA9, offering new insights into the processes involved. Consequently, this research study not only underscores the biostimulant potential of DOA9 but also illuminates the defense mechanisms activated against CLS through priming with the DOA9 strain.

## 2. Results

### 2.1. Bradyrhizobium sp. DOA9-Mediated Priming Triggers Brown Spot Resistance in Mung Bean

To investigate the potential of bio-priming mediated by DOA9 inoculation, the plants’ roots were inoculated with DOA9 before evaluating the lesion size of the brown spot disease caused by *Cercospora canescens* infection. Plants were inoculated with DOA9 for 17 days (LP) or 3 days (SP) and for both inoculation times (DP) before being infected with *C. canescens* (+CC) (Figure 1a). The phenotype of plants subjected to DOA9 priming did not differ from that of non-inoculated (NI) plants because DOA9 did not enhance plant growth. Thus, this experiment included a nitrogen source in the plant medium to ensure consistent growth across all treatments. The results showed that plants without priming (NI+CC) and those with LP (LP+CC) exhibited larger brown spot disease infections than those with SP (SP+CC) and DP with DOA9 (DP+CC) (Figure 1b). This finding aligns with the results of the lesion size evaluation using the ImageJ program version 1.54 f (http://imagej.net/Fiji, accessed on 14 September 2019) (Figure 1c). At 1 day post fungal inoculation (dpi), a significant reduction in lesion size of about 49% and 50% was observed under SP+CC and DP+CC treatments, respectively, compared to NI+CC. Similarly, at 2 dpi, a smaller lesion size—approximately 80% and 81% reduction—was noted under SP+CC and DP+CC treatments, respectively, compared to NI+CC. In contrast, lesion size was slightly reduced under the LP+CC treatment but not significantly different from NI+CC. These results suggest that both short and double priming with DOA9 are effective strategies to reduce the severity of leaf spot disease, whereas without DOA9 priming or long priming, it might not be helpful.

Interestingly, there was no significant difference observed in the *C. canescens* copy number (*TEF-1α*) across all treatments (Appendix A). However, the pathogen’s copy number significantly dwindled after 3 days of infection (Appendix A). This indicates that during the initial stage of infection, fungal growth and replication might be constrained. Remarkably, the copy number did not change significantly on days 2, 3, 7, and 10 dpi for any treatment groups (Appendix A). This seems to conflict with the resistance phenotype and lesion size data, where lesion size amplified in the NI+CC and LP+CC treatments, but the SP+CC and DP+CC treatments showed diminished lesion formation. This implies that the short and double priming treatments may not directly restrict fungal growth. Instead, they might target specific virulence factors of the pathogen, prevent infection, or lessen disease severity without entirely eradicating the fungal population.

To test these hypotheses, fungal colonization in the infection zone of each treatment group was initially investigated 2 and 10 days after inoculation with *C. canescens* (Appendix A). The results displayed no significant disparities in either mycelium density or phenotype between the treatments. Mycelia were seen to grow and expand within the infection zones. These findings support the hypothesis that both short and double priming with DOA9 on roots does not directly affect fungal growth. However, the treatments may induce chemical changes related to the plant defense reaction in plant cells, leading to reduced disease severity without complete fungal elimination. To gain more insight into the mechanisms underlying the role of plant defense responses mediated by DOA9 priming, gene expression in leaves with and without pathogen infection was investigated.

### 2.2. Bio-Priming by Bradyrhizobium sp. DOA9 Triggers Plant Immunity

To elucidate the mechanisms of plant immunity induced by DOA9 bio-priming on roots and *C. canescens* infection on leaves, we analyzed the expression profiles of genes involved in the SA and JA signaling defense pathways (Figure 2a) at 1 and 2 dpi. At 1 dpi, the gene expression profile of *V. radiata* CN72 exhibited a wide range of responses to DOA9 root priming and *C. canescens* infection, with high expression levels observed under *C. canescens* treatments. However, at 2 dpi, the heatmap indicated that genes were most up-regulated in plants subjected to both DOA9 priming and *C. canescens* infection, suggesting a synergistic effect on the activation of defense-related genes (Figure 2b).

At 1 dpi, *Pti1* expression exhibited significant up-regulation under SP compared to the NI control and similarly under short priming with *C. canescens* infection (SP+CC) compared to the NI control with *C. canescens* infection (NI+CC) (Figure 3a). In contrast, *PR-1* was dramatically overexpressed under DP with *C. canescens* infection (DP+CC), while its underexpression was observed in other treatments (Figure 3b). *PR-2* expression could be induced by the three treatments of DOA9 priming, NI+CC, and SP+CC, but remained unchanged in *C. canescens*-infected mung bean leaf under LP with *C. canescens* infection (LP+CC) and DP+CC treatments compared to NI (Figure 3c). Notably, *PR-5* was suppressed under LP, NI+CC, LP+CC, and DP+CC compared to NI, while SP, DP, and SP+CC maintained *PR-5* expression as found in NI (Figure 3d). *Prx* expression remained unchanged under DOA9 priming, but it significantly increased under *C. canescens* infection treatments (Figure 3e). The *HR* gene did not change under any DOA9 priming, but it showed a slight increase in all *C. canescens*-infected treatments (Figure 3f).

Additionally, *NDR1* expression remained constant with DOA9 priming alone. However, the highest expression of *NDR1* was observed under SP+CC. Notably, high expression levels were also prominent under *C. canescens* infection alone. On the contrary, *NDR1* expression was reduced under DP+CC compared to NI+CC (Appendix A). The expression of *EDS1* maintained consistency with DOA9 priming relative to NI but was up-regulated under LP+CC in comparison to NI+CC. *Pti-5* expression was notably unaffected by DOA9 priming but was reduced under *C. canescens* infection. *Pti-6* expression was significantly reduced under SP and DP and was also diminished under *C. canescens* infection relative to NI. *PR-3* expression remained without significant changes. *PR-4* expression decreased under LP and dramatically increased under *C. canescens* infection; however, it was significantly reduced under SP+CC and DP+CC relative to NI+CC. *CHS* expression remained stable with DOA9 priming relative to NI but was notably expressed under *C. canescens* infection, with significant reductions under LP+CC and SP+CC in comparison to NI+CC (Appendix A). These results suggest that DOA9 priming escalates the expression of *Pti1* and *PR-2* and triggers a higher expression of *Pti1*, *PR-1*, *PR-5*, *Pti6*, and *EDS1* in *C. canescens*-infected plants compared to NI+CC. Subsequently, the gene expression pattern during *C. canescens* infection was examined at 2 dpi.

At 2 dpi, *Pti1* expression in *V. radiata* CN72 was notably up-regulated under all DOA9 priming treatments compared to NI. Interestingly, the maximum *Pti1* expression was observed under NI+CC. However, the expression of *Pti1* was significantly reduced under LP+CC, SP+CC, and DP+CC compared to NI+CC (Figure 3g). This pattern mirrored the expression of the *HR*, *PR-3*, and *NDR1* genes (Appendix A).

The expression of *Pti5* significantly increased under LP and DP conditions when compared to NI, presenting the highest expression levels in most *C. canescens*-infected treatments, particularly under LP+CC (Figure 3h). *EDS1* also showed up-regulation under both LP and SP compared to NI. Moreover, *EDS1* was significantly up-regulated in the DP+CC treatment in comparison to the NI+CC treatment (Figure 3i).

The expression of *CSH* was significantly up-regulated under LP and DP, with high expression levels of *CHS* observed in plants after *C. canescens* infection (Figure 3j). Notably, the expression of *Prx* remained unaltered by DOA9 priming alone but showed a significant increase in all DOA9-primed plants after *C. canescens* infection (LP+CC, SP+CC, and DP+CC) (Figure 3k). This outcome exhibited an inverse expression pattern compared to *Pti1*, *HR PR-3*, and *NDR1*. Additionally, *C. canescens* infection remarkably increased the expression of the genes *Pti1*, *HR*, *PR-3*, and *NDR1*, yet DOA9 priming stifled these genes’ expression levels. On the other hand, *Prx* expression notably surged in DOA9-primed plants following *C. canescens* infection (Figure 3g–i,k,l). The interpretation of these results suggests that DOA9 priming could possibly defend against fungal infection by inducing the expression of *Prx*. *Prx* encodes peroxidase enzyme which abates oxidative damage during pathogen infection while preventing the excessive activation of the HR, which is capable of causing oxidative damage in mung bean.

The expression of *Pti6* in *V. radiata* CN72 was up-regulated under LP conditions and experienced a slight increase under SP and DP compared to NI. When infected with *C. canescens*, *Pti6* expression significantly accumulated under LP+CC and SP+CC conditions compared to NI+CC. The expression levels of *PR-2* and *PR-4* remained unchanged under DOA9 priming relative to NI, but higher levels were revealed under *C. canescens* infection. As for *PR-5*, its expression did not change with DOA9 priming compared to NI and showed similar levels under *C. canescens* infection. However, it was significantly up-regulated under LP+CC compared to NI+CC (Appendix A).

Altogether, this implies that priming with DOA9, particularly LP, enhances the plant’s defense mechanisms by up-regulating key defense-related genes such as *Pti6* and *PR-5*, particularly in the presence of a pathogen. Moreover, the significant increase in *Prx* expression in all DOA9-primed plants following *C. canescens* infection indicates that priming enhances the plant’s ability to detoxify reactive oxygen species (ROS) and mitigate oxidative damage during a pathogen attack.

### 2.3. Hydrogen Peroxide and Phenolic Content and Enzyme Activities

The investigation of secondary metabolites in response to DOA9 priming and *C. canescens* infection in mung beans involved the evaluation of hydrogen peroxide (H_2_O_2_) and total phenolic content. Enzyme activity, including peroxidase (POD), superoxide dismutase (SOD), chitinase, and *β*-glucanase activities, was also assessed at 1 and 2 dpi. POD activity significantly increased under LP, SP, and DP compared to NI. Similarly, plants primed with DOA9 and infected by *C. canescens* showed a substantial increase in POD activity (Figure 4a,g). However, SOD, chitinase, and *β*-glucanase activities, as well as the phenolic content, showed no significant changes at 1 and 2 dpi (Figure 4b–d,h–j). The H_2_O_2_ content under the LP, SP, and DP treatments was similar to that of NI. Yet, at 1 dpi, H_2_O_2_ levels were significantly higher under the LP+CC and SP+CC treatments compared to NI+CC. By 2 dpi, H_2_O_2_ content had decreased under SP compared to NI and was lower under LP+CC, SP+CC, and DP+CC compared to NI+CC (Figure 4e,k). These results suggest that DOA9 priming and *C. canescens* infection may influence the accumulation of POD and H_2_O_2_ production in *V. radiata* CN72 leaves (Figure 4).

### 2.4. The Impact of Bradyrhizobium sp. DOA9 Exopolysaccharide (EPS) on Induced Resistance of Plants against C. canescens

Given the potential of DOA9 priming to enhance resistance against CLS, we hypothesize that exopolysaccharides (EPS), which represent one of the PAMPs, may instigate a basal defense response in plants. As such, this study explores the impact of *Bradyrhizobium* sp. DOA9 EPS on inducing plant resistance to *C. canescens*. In the experiment, plants were inoculated with DOA9 (DOA9-DP) and the extracted EPS (EPS-DP), employing a double priming method, before being infected with *C. canescens* (Figure 5a).

Large brown spots and significant lesion sizes were observed in the NI control with *C. canescens* infection (NI+CC) and the EPS-treated DP with *C. canescens* infection (EPS-DP+CC). In contrast, the DP with *C. canescens* infection (DP+CC) showed fewer brown spots and smaller lesion sizes (Figure 5b,c). These results suggest that EPS may not effectively induce resistance in mung bean against *C. canescens*. Thus, it is interesting to further investigate the role of T3SS, which may be involved in triggering this resistance.

### 2.5. The Impact of Bradyrhizobium sp. Type 3 Secretion System (T3SS) on Resistance of Plants against C. canescens

The effector proteins secreted by the Type 3 Secretion System (T3SS) of bacterial pathogens or rhizobia can trigger the plant defense system referred to as ETI. This defense strategy involves a powerful response, including localized cell death, which is known as a HR. In addition to the Nod-factor signaling pathway, *Bradyrhizobium* sp. DOA9 has a T3SS that plays a critical role in its symbiotic relationship with *V. radiata* [35] (Figure 6a). To investigate the role of the T3SS in plant resistance against *C. canescens*, we conducted a double-priming experiment using a T3SS-deficient mutant of DOA9, designated as Ω*rhcN*. This mutant strain has a disrupted *rhcN* gene, which encodes a crucial ATPase necessary for the proper function of the T3SS injectisome. Unlike the wild-type DOA9, Ω*rhcN* is unable to secrete type 3 effector proteins into plant cells, leading to the formation of effective nodules (Figure 6a). The double-priming experiment was performed in comparison with the wild-type DOA9 strain (Figure 6b). Next, we evaluated the size of the lesions caused by brown spot disease after *C. canescens* infection. The results were compared with those from plants primed with the DOA9 wild-type strain (Figure 6b).

The largest brown spots and lesion sizes were observed under NI+CC but were significantly reduced under the Ω*rhcN*-DP+CC and DP+CC treatments. However, compared to the DP+CC treatment, brown spots and lesion sizes significantly increased with the Ω*rhcN*-DP+CC treatment (Figure 6c,d). Additionally, small and necrotic nodules were found under DP+CC, whereas normal-appearing nodules were observed under Ω*rhcN*-DP+CC (Figure 6a). These results demonstrated that the T3SS plays a negative role in symbiosis, but it may still partly contribute positively to the induction of resistance in *V. radiata* CN72.

## 3. Discussion

*C. canescens* poses a threat to mung beans, causing CLS, which can lead to a reduction in productivity. Lesions on leaves can result from cercosporin produced by *C. canescens* [37] or from the plant’s HR to pathogen infection [38], both of which are often associated with ROS production. Our results revealed that SP and DP with DOA9 were effective in lessening the severity of leaf spot disease by diminishing lesion size on *V. radiata* CN72 (Figure 1). However, the number of *TEF-1α* copies and hyperpopulation detected on leaves remained similar across all treatment groups (Appendix A). This suggests that short and double priming may not directly inhibit fungal growth but could help alleviate the damage inflicted on *V. radiata* CN72 leaves through other mechanisms.

Bio-priming has been shown to significantly reduce disease incidence and severity in various crops by enhancing their innate immune responses [39]. This study examined the expression of plant defense-related genes involved in the SA signaling pathway, including the NDR1/HIN1-like protein 10 (*NDR1*), *EDS1* protein, and the pathogenesis-related genes transcriptional activators *Pti5* and *Pti6*. All these genes act upstream of the SA signaling pathway to regulate SA accumulation [40,41]. *NDR1* has also been implicated in HR during incompatible plant–pathogen interactions in an SA-dependent manner [42]. From our results, a high expression of *NDR1* was detected in SP+CC at 1 dpi (Appendix A). The expression of *NDR1* was further up-regulated at 2 dpi in plants infected with *C. canescens*, while DOA9-primed plants showed down-regulation compared to the NI+CC (Appendix A). This suggests that *NDR1* responds to pathogen infection, but DOA9 priming might act as a negative regulator.

In our study, *EDS1* is crucial for the activation of SA-dependent defense responses, including the expression of *PR* genes (*PR1*, *PR2*, *PR5*) [25]. The transcriptional regulators *Pti5* and *Pti6* are modulated via the SA and JA/ET signaling pathways [43] and are players in the intricate network of plant defense by activating a broad spectrum of PR genes such as *PR-1*, *PR-2*, and *PR-5* [44]. *EDS1* expression was found at 2 dpi under LP and SP conditions (Figure 3i, Appendix A). However, this gene began its LP+CC expression at 1 dpi (Appendix A). Subsequently, the expression of *EDS1* increased under LP, SP, NI+CC, SP+CC, and DP+CC at 2 dpi (Figure 3i, Appendix A). The response of *EDS1*-like expression is involved in grapevine-resistant responses against *Rhizhobium vitis* [45]. Also, the HR-inducing gene enhances plant basal resistance through an EDS1 and SA-dependent pathway [46]. Thus, the expedited *EDS1* expression under LP+CC might involve *HR*, as evidenced by its expression at 1 dpi, which consequently causes lesions on leaves at 1 dpi.

The expression patterns of transcriptional regulators *Pti5* and *Pti6* revealed initial suppression by *C. canescens* infection (Appendix A), suggesting the pathogen may interfere with the plant’s initial defense mechanisms. However, we noted the subsequent induction of *Pti5* and *Pti6* at 2 dpi, particularly under priming treatments (Appendix A). This suggests the plant’s defense system eventually prevailed over the pathogen’s suppression, triggering a robust defense response. These findings align with prior studies on tomatoes that demonstrated the overexpression of *Pti5* and *Pti6* amplifies pathogen-induced defense gene expression and heightens resistance to *Pseudomonas syringae* pv. tomato [44].

High expression of the *PR-1* gene was found under DP+CC at 1 dpi (Figure 3b, Appendix A), but it remained unchanged in other treatments and was not detected in any treatments at 2 dpi (Appendix A). *PR-1* is a well-known marker of the SA signaling pathway, which encodes an antifungal peptide [47]. It plays a crucial role in modulating plant metabolism in response to both biotic and abiotic stresses such as *Fusarium oxysporum* [48]. The beneficial function of *GmPR1L* in triggering *Glycine max* resistance against *C. sojina* infection has been demonstrated [49]. Therefore, this result indicates that high expression under DP+CC may confer resistance to *C. canescens* invasion.

DOA9 priming was also able to induce *PR-2* expression in all treatments at 1 dpi, but it remained unchanged at 2 dpi (Appendix A). The highest expression of *PR-2* was found in SP+CC at 1dpi, compared to NI+CC. *PR-2* is a crucial component of the plant’s antifungal defense mechanisms (*β*-1,3-*glu*canase), helping to limit pathogen spread and infection [50]. It has been reported that *β*-1,3-glucanase exhibits varying inhibitory effects on the hyphal growth of *F. graminearum*, *Alternaria* sp., *Aspergillus glaucus*, *A. flavus*, *A. niger*, and *Penicillium* sp. in vitro [51]. These results suggest that *PR-2* expression in *V. radiata* CN72 was enhanced by DOA9 priming, particularly SP, and it also responded to *C. canescens* invasion. However, the gene expression level and the corresponding *β*-1,3-glucanase activity were not congruent (Appendix A, and Figure 4). These results imply that there is no direct correlation between the mRNA expression levels of *β*-1,3-glucanase genes and the actual enzymatic activity of the *β*-1,3-glucanase protein.

The suppression of *PR-5* was observed under LP, NI+CC, LP+CC, and DP+CC at 1 dpi (Figure 3d, Appendix A) and under NI+CC at 2 dpi (Appendix A). The *PR-5* family, which includes thaumatin-like proteins (TLPs), plays a crucial role in facilitating resistance against both abiotic and biotic stress responses [52]. Rather than directly eradicating the pathogens, TLPs function as signal/elicitor factors, bolstering plant resistance to fungal pathogens [53]. *PR-5* can inhibit hyphal growth and spore germination via the formation of transmembrane pores, which result in fungal cell leakiness and obstruct the functionality of molecules that serve as plasma membrane receptors in the cAMP/RAS2 signaling pathways [54]. The thaumatin-like protein has shown impressive effectiveness in conferring resistance against fungal pathogens, salt stress, and oxidative stress in both in vivo and in vitro environments [55]. However, rhizobia must possess unique characteristics to dodge and temporarily inhibit plant defense responses, facilitating their penetration into root tissues. These traits are vital for successful colonization and symbiosis with the host plant [56]. Additionally, the fungal pathogen delivers effectors into the host plant cells to weaken the host’s immune responses and establish infection [57]. Thus, the lack of *PR-5* expression induced by LP, NI+CC, LP+CC, and DP+CC may be a strategy of evasion used by DOA9 and *C. canescens*, causing suppression of plant immunity. This could negatively impact plant resistance to *C. canescens* invasion.

In the case of the JA/ET signaling pathway, this study examined the expression of *PR-3*, which encodes chitinase, and *PR-4*, encoding pathogenesis-related protein PR-4. These serve as indicators of JA-dependent SAR [58]. The chalcone synthase (*CHS*) is a crucial enzyme in the flavonoid/isoflavonoid biosynthesis pathway, regulated by JA [59,60,61]. Peroxidase (POD) participates in lignification, increasing the resistance capabilities of the cell wall by generating mechanical pressure during fungal attacks and penetration [62]. The *Pti1* gene encodes a serine/threonine kinase (protein kinase) involved in the HR [63,64] and is also implicated in the expression of ROS [65]. Moreover, *Pti1* mediates the rise in endogenous H_2_O_2_ levels. Endogenous H_2_O_2_ is associated with the process of lignification [66] (Figure 2a).

In this study, *Prx* expression remained unchanged by DOA9 priming in *V. radiata* CN72 at 1 and 2 dpi (Figure 3e,k). However, POD activity in all primed plants was significantly increased compared to the NI control, especially at 2 dpi (Figure 4). This result corresponds to the expression level at 2 dpi when plants were primed with DOA9 and infected with *C. canescens*. On the other hand, the accumulation of the peroxidase *rip1* transcript was rapidly and transiently induced by *R. meliloti*, then declined by 48 h. This expression pattern coincided with the pre-infection period and early infection events, aligning with the onset of nodule morphogenesis [67]. In response to the *C. canescens* infection, the plant host may induce the expression of pathogen-inducible peroxidase enzymes that play a role in defense against phytopathogens.

Interestingly, LP+CC, SP+CC, and DP+CC induced a more significant increase in *Prx* expression than NI+CC. Therefore, DOA9 priming might enhance *Prx* expression when hosts are invaded by *C. canescens*. *Pti1* expression in *V. radiata* CN72 under SP and SP+CC was higher than NI+CC at 1 dpi (Figure 3a), indicating that SP enhances *Pti1* expression early in infection. However, at 2 dpi, the expression of *Pti1* was induced in all plants treated with DOA9 priming and *C. canescens* infection, yet its expression was lower in plants with DOA9 priming and *C. canescens* infection than in NI+CC. *Pti1* activation under SP and SP+CC at 1 dpi may trigger immune responses, such as producing ROS as H_2_O_2_, which can inhibit pathogen infection and facilitate other defense responses, such as lignification, ultimately suppressing the pathogen in *V. radiata* CN72 early in the infection. It seems that DOA9 priming might enhance *Prx* expression when hosts are invaded by *C. canescens*.

*HR* expression was observed to be induced by DOA9 priming and *C. canescens* infection at 1 dpi compared to NI. *HR* expression was particularly high under NI+CC and LP+CC conditions, potentially leading to larger lesions on *V. radiata* CN72 under these conditions as compared to SP+CC and DP+CC. This suggests that DOA9 priming might suppress *HR* expression. By 2 dpi, LP had activated *HR* expression. A previous study reported that *B. elkanii* T3SS up-regulated soybean genes, including defense-related genes encoding the HR protein. Although *HR* expression was high under NI+CC, it was significantly reduced under LP+CC, SP+CC, and DP+CC (Figure 3l). This finding aligns with H_2_O_2_ content, which was reduced in primed plants (Figure 4). *HR* is a crucial component of the plant immune system, playing a key role in restricting the spread of pathogenic infections and leading to programmed cell death (PCD) at the site of the infection [16,19,68]. *HR* can be triggered through various pathways and might manage the plant defense responses in both local and distant tissues [38]. Considering our findings, *HR* expression may correlate with lesion size, with the largest observed under NI+CC followed by LP+CC and the smallest under SP+CC and DP+CC.

The expression of *PR-3* appears to have a slight response at 1 dpi, whereas *PR-4* was highly expressed in *C. canescens*-infected plants (Appendix A). DOA9 priming did not stimulate *PR-4* expression, but it might have been affected by the reduction in *PR-4* expression found under LP, SP+CC, and DP+CC at 1 dpi. However, at 2 dpi, *PR-4* was highly expressed in *C. canescens*-infected plants (Appendix A). PR-4 protein in *Malus domestica* (*MdPR-4*) has been reported to inhibit hyphal growth during *Botryosphaeria dothidea* infection [69]. Furthermore, fungal infection tends to induce *PR-4* expression in maize, potentially triggering a general defense response against pathogens [70]. These results suggest that the activation of *PR-3* and *PR-4* expressions might be a mechanism of plant response to *C. canescens* infection.

Additionally, we explored the possibility of certain factors from DOA9 inducing the defensive response in *V. radiata* CN72, thereby enhancing resistance to *C. canescens* with a specific focus on EPS and T3SS. The results showed that EPS priming in roots did not affect the resistance of *V. radiata* CN72 to *C. canescens,* as shown by the unaltered lesion size on leaves. However, when primed with DOA9 lacking T3SS, resistance notably decreased, a fact attested by the enlarged lesion sizes on leaves. Previous research has shown that type 3 effectors (T3Es) of DOA9 secreted via T3SS can stimulate a plant’s immune response [36]. These findings suggest that T3SS plays a pivotal role in enhancing the resistance of *V. radiata* CN72 to *C. canescens*. Nonetheless, the identification of specific effector proteins secreted by T3SS-DOA9 that contribute to the priming system is earmarked for future investigation. This research advances our understanding of developing potentially efficacious biostimulants, providing an alternative to the use of chemical treatments in agriculture.

The findings demonstrate that root priming with *Bradyrhizobium* sp. DOA9 can effectively enhance resistance traits against CLS in mung bean. However, DOA9 priming does not directly inhibit fungal growth but rather strengthens the plant’s physical and biochemical defenses. The gene expression and enzyme activity results suggest a schematic overview of plant defense reactions mediated by DOA9 priming (Figure 7). In the roots, DOA9 priming and *C. canescens* infection on the leaves of *V. radiata* CN72 can induce or suppress genes related to defense. Notably, the resistance traits of *V. radiata* CN72 to *C. canescens* infection were observed under SP+CC and DP+CC treatments. Therefore, genes activated by SP, including *Pti1*, *EDS1*, and *PR-2*, and by DP, including *Pti1*, *Pti5*, *PR-2*, and *CHS*, as well as an increase in POD activity, might significantly contribute to resistance prior to pathogen invasion. The activation possibly occurs in the roots through T3Es via the T3SS of DOA9, transmitting the signal to the leaves. The elevated expression of *Pti1*, *Pti6*, *NDR1*, *Prx*, and *PR-5*, observed due to being unsuppressed under SP+CC and *EDS1*, *PR-1*, and *Prx* under DP+CC, may play a significant role in *C. canescens* resistance (Figure 7).

## 4. Materials and Methods

### 4.1. Plant Materials

The mung bean CN72 (*Vigna radiata*) was utilized in this experiment. The seeds were surface-sterilized by being soaked in 95% EtOH for 1 min, followed by being washed with sterilized water three times. Subsequently, the seeds were treated in 3% (*v*/*v*) NaOCl for 5 min, washed with sterilized water three times, and then soaked in sterilized water for 12 h. The sterilized seeds were germinated on 1% (*w*/*v*) water agar and then incubated in darkness for 1 day. The germinated seeds (two plants per pot) were planted in Leonard’s jars (383 cm^3^) containing sterilized vermiculite. BNM medium [71] was added to provide plant nutrients, and the jars were placed in a light room maintained at 28 ± 2 °C. The room followed a 12/12 h day/night regimen at 300 μE m^−2^ S^−1^ light intensity with 50% humidity. After 14 days, the BNM medium was replaced with BNM + 2 mM KNO_3_ medium until the result was recorded.

### 4.2. Preparation of Bradyrhizobium sp. Strain DOA9 Inoculum, Cercospora Canescens, and Exopolysaccharide (EPS) Production

The *Bradyrhizobium* sp. strain DOA9 and its derivative Ω*rhcN* mutant used in this study are listed in Table 1. The DOA9 wild-type strain and Ω*rhcN* were cultured in YEM medium with antibiotic supplementation (200 µg/mL of spectinomycin) [35]. They were then collected using centrifugation (4000 rpm, 20 min) and subsequently washed twice with sterilized 0.85% (*w*/*v*) NaCl. The cell concentration was adjusted at OD_600_ nm to 1.0 (10^8^ CFU mL^−1^) prior to being used as inoculum. In preparation for the inoculum of the pathogenic fungus responsible for CLS, *Cercospora canescens* strain PAK1′s mycelial suspension was obtained, referencing a protocol previously described by [72]. To yield EPS from DOA9, a 1% (*v*/*v*) starter culture was inoculated into 500 mL of liquid YEM medium and incubated at 30 °C (150 rpm) to achieve 10^8^ CFU mL^−1^. The total EPS was extracted and conditioned as previously described [73]. The resulting EPS powder was then dissolved in 500 mL of sterilized DI water, ready for EPS inoculation.

### 4.3. Experimental Design

Seven-day-old mung beans in controlled environmental conditions were divided into two groups: (i) *C. canescens*-infected (+CC) and (ii) non-infected (−CC) plants. Each group was further divided into four treatments, including (I) non-DOA9-inoculated plants used as controls (NI); (II) plants with single inoculation by DOA9 at 5 days after planting (or 17 days before fungal infection), referred to as long priming (LP); (III) plants with single inoculation by DOA9, 3 days before fungal infection, referred to as short priming (SP); and (IV) plants with double inoculation by DOA9, 17 and 3 days before fungal infection, referred to as double priming (DP). So, in total, there were eight treatments: four without fungal infection (NI, LP, SP, and DP) and four with fungal infection (NI+CC, LP+CC, SP+CC, and DP+CC). The inoculation of DOA9 cells was carried out by applying a 1 mL suspension to the root of each plant. For leaf infestation with pathogenic fungi, six drops of 5 µL of *C. canescens* mycelial suspension were placed on the underside of the second leaf node from the top. To maintain conditions conducive to disease development, the plants were housed within closed containers [74]. The experiments involved four biological replicates for each treatment, with at least 24 individual leaf infections evaluated per replicate. Following fungal infection, lesion size caused by *C. canescens* was assessed at 1 and 2 days post-infection (dpi) using ImageJ program version 1.54f (National Institutes of Health, Bethesda, MD, USA).

To evaluate the potential of EPS and the impact of DOA9′s T3SS on plant resistance against CLS, DP was performed on plants with either EPS or cells of the derivative T3SS mutant (Ω*rhcN*) in comparison to the DOA9 wild type (1 mL of inoculation was conducted in the same manner for priming as previously described). After fungal infection, lesion size and resistance phenotype were evaluated at 2 dpi, as mentioned previously, by comparison with non-inoculated (NI) plants.

### 4.4. Evaluation of C. canescens DNA Copy Number and Hyphae Colonization on Infected Leaves

The *TEF-1α* gene was utilized as a genetic marker to gauge the quantity of *C. canescens* in mung bean leaves at 2, 3, 7, and 10 dpi. A total of 0.5 g of infected leaves was employed for DNA extraction using the CTAB method. Briefly, 100 mg of powder was added to 700 µL of lysis buffer (containing 2% CATB buffer and 3% Polyvinylpyrrolidone (PVP)), along with 20 µL of 10% Sodium Dodecyl Sulfate (SDS), and then incubated at 65 °C for 30 min. Subsequently, 700 µL of chloroform was added and mixed before centrifuging at 12,000 rpm for 10 min. The aqueous phase was transferred to a new tube and mixed with 700 µL of chloroform prior to another centrifugation at 12,000 rpm for 10 min. DNA was then precipitated with 1.5 (*v*/*v*) of cool absolute ethanol. After being washed with 70% (*v*/*v*) ethanol and dried, the DNA was re-dissolved in nuclease-free water. The DNA sample was purified using the FavorPrep™ Tissue Genomic DNA Extraction Kit (Favorgen, Ping-Tung, Taiwan). The total DNA concentration was adjusted to 5 ng/µL and used for the evaluation of the *TEF-1α* gene copy number via quantitative PCR analysis (qPCR) using specific primers (TEF-RT.F; 5′TGGACACCACCAAGTGG3′ and TEF-RT.R; 5′GGTCTTGGTCTCCTTCT3′), designed from the translation elongation factor 1α (accession no. MN086775.1). The qRT-PCR reaction was conducted using the Luna^®^ Universal qPCR Master Mix (NEB, Boston, MA, USA) and performed on the Bio-Rad CFX Opus 96 Real-Time PCR System (Bio-Rad, Singapore). The copy number was analyzed by comparing to the standard curve in each 10-fold dilution that housed a known copy number of pTG19-T (2880 bp) vector containing the *TEF-1α* (191 bp) gene at 3.017 × 10^7^ to 3.017 × 10^3^ copies/µL. The concentration in each dilution was converted to copy numbers using an equation as described by [75]. The fungal hyphae colonization on leaves was examined at 2 and 10 dpi. Infected leaves were collected and stained by boiling them for 2 min in an alcoholic lactophenol trypan blue solution, as described by [76]. The leaves were then soaked in lactoglycerol (1:1:1 lactic acid/glycerol/water) to de-stain overnight, and the hyphae were observed under a light microscope.

### 4.5. Total RNA Extraction and qRT-PCR Analysis

The leaves with and without fungi infection were collected 1 and 2 days after *C. canescens* infection for RNA extraction. A total of 100 mg of leaf powder was extracted using the FavorPrep Plant Total RNA Purification Mini Kit, following the manufacturer’s protocol (Favorgen). A total of 500 ng of total RNA was converted to cDNA using the iScript^TM^ cDNA Synthesis Kit (Bio-Rad). The qRT-PCR analysis was performed using the Luna^®^ Universal qPCR Master Mix (NEB, Boston, MA, USA), with specific primers (Table 1) which were designed from database of *V. radiata* var. *radiata* (mung bean) (Ref Seq assembly accession: GCF_000741045.1). Relative gene expression was calculated using the comparative Ct method (−ΔΔCT), compared with the endogenous housekeeping gene Actin [77].

### 4.6. Determination of Enzyme Activities and Total H_2_O_2_ and Phenolic Compound Contents

The activities of antioxidant enzymes, including superoxide dismutase (SOD) and peroxidase (POD), as well as total H_2_O_2_ and phenolic compound contents, were measured in leaves on days 1 and 2 after fungal infection compared to non-infected plants. Plant leaves were ground with liquid nitrogen and used for further analysis. The total H_2_O_2_ and phenolic compound contents and SOD enzyme activity were measured using the method described by [78], while the POD enzyme activity was determined using the method described by [79].

The protocol described by [80] was used to detect *β*-1,3-glucanase and chitinase activities in infected leaves (on days 1 and 2) and non-infected leaves from all treatments. *β*-1,3-glucanase and chitinase activities were calculated based on the total protein concentration, which was detected using the Quick Star Bradford Protein Assay (Bio-Rad) with bovine serum albumin (BSA) as a standard.

### 4.7. Statistical Analysis

The mean values and standard error were analyzed using SPSS software (SPSS version 26.0 Windows; SPSS Inc., Chicago, IL, USA). Comparisons of the means were conducted using an independent sample *t*-test at *p* < 0.05 for related gene expression and at *p* < 0.001 for lesion size. The results were expressed as the mean and standard error of the mean (SEM).

## 5. Conclusions

This study highlights the potential of DOA9 as a bio-priming agent to enhance the resistance in *V. radiata* CN72 against *C. canescens*, the causative agent of CLS. The results showcase that short and double priming with DOA9 significantly reduces the size of the lesions on infected leaves by activating or amplifying defense-related genes such as *Pti1*, *Pti6*, *EDS1*, *NDR1*, *PR-1*, *PR-2*, *Prx*, and *CHS*, in addition to suppressing the inhibition of *PR-5*. Notably, the DOA9-T3SS may be partially implicated in the resistance of *V. radiata* CN72. This suggests that DOA9 effectively primes the defense mechanisms of *V. radiata* CN72, providing a sustainable alternative to chemical fungicides for managing CLS.

## Figures and Tables

**Figure 1 plants-13-02495-f001:**
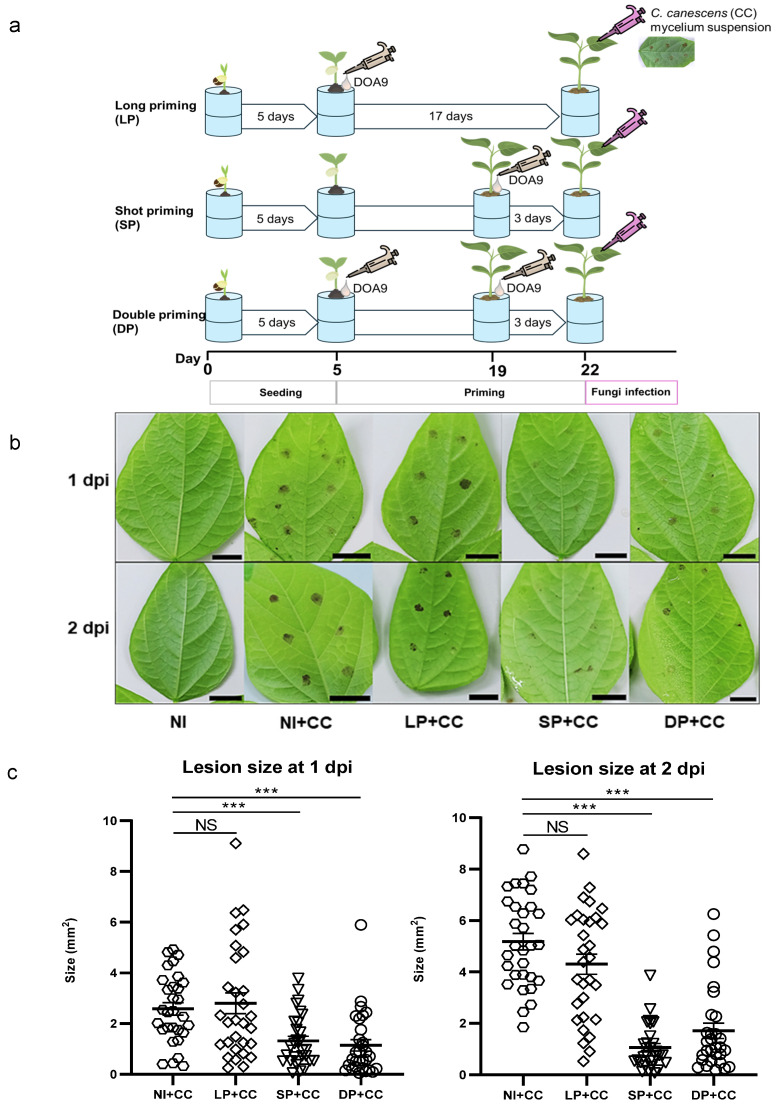
Phenotype of Cercospora leaf spot (CLS) symptoms in *V. radiata* cv. CN72 in response to bio-priming using *Bradyrhizobium* sp. DOA9 inoculum. The time schedule of seeding, DOA9 priming in different strategies, including long priming (LP), short priming (SP), and double priming (DP), and fungi infection by *C. canescens* (+CC) are explained (**a**). CLS lesion on the leaf after infection with *C. canescens* (+CC) at 1 and 2 days post-infection (dpi) (**b**); non-DOA9-inoculated plant was used as a control (NI); non-DOA9-inoculated plant with *C. canescens* infection (NI+CC); long priming with *C. canescens* infection (LP+CC), short priming with *C. canescens* infection (SP+CC)), and double priming *C. canescens* infection (DP+CC) (black scale bar indicates centimeters (cm)). Lesion sizes measured by ImageJ were presented in dot plot (*n* = 30) (**c**). Data represent mean ± SEM. Symbols indicate statistical significance (*t*-test *** *p* < 0.001; NS = not significant).

**Figure 2 plants-13-02495-f002:**
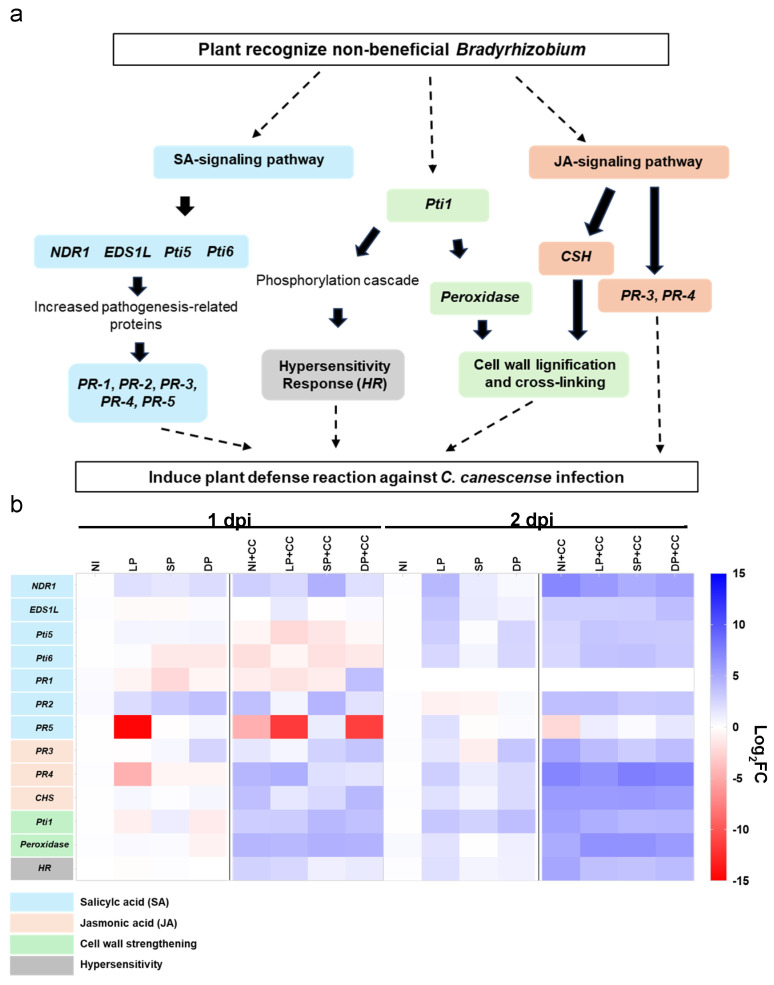
Relative gene expression of *V. radiata* CN72. (**a**) Schematic representation of genes related to plant defense. Solid arrows indicate known pathway and dotted arrows indicate unknown pathway (**b**) Heatmap showing the Log_2_ fold change (Log_2_FC) of relative gene expression in *V. radiata* CN72 leaves at 1 and 2 dpi with the fungal pathogen *C. canescens*. Non-DOA9-inoculated plant was used as a control (NI). Treatment methods: long-priming (LP), short-priming (SP), and double-priming (DP), non-DOA9-inoculated plant with *C. canescens* infection (NI+CC), long priming with *C. canescens* infection (LP+CC), short-priming with *C. canescens* infection (SP+CC), and double-priming *C. canescens* infection (DP+CC). Gene expression in leaves infected with *C. canescens* was compared to non-infected controls. The genes involved in the plant immune response included *NDR1* (NDR1/HIN1-like protein 10), *EDS1* (Protein EDS1), *Pti5* (Pathogenesis-related genes transcriptional activator Pti5), *Pti6* (Pathogenesis-related genes transcriptional activator Pti6-like), *PR-1* (Pathogenesis-related protein 1), *PR-2* (*β*-1,3-glucanases), *PR-5* (Thaumatin-like protein), *PR-3* (Chitinase 10), *PR-4* (Pathogenesis-related protein 4), *CHS* (Chalcone synthase 17-like), *Pti1* (Pti1-like tyrosine-protein kinase At3g15890), *Prx* (Peroxidase 10), and *HR* (Hypersensitive-induced response protein 2-like). Data represent mean ± SEM; *n*= 3.

**Figure 3 plants-13-02495-f003:**
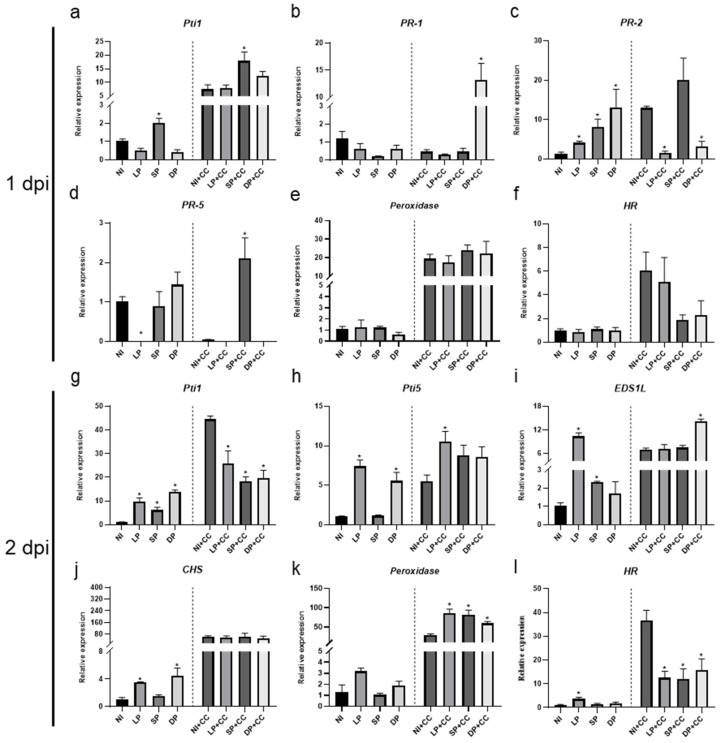
The relative gene expression in *V. radiata* CN72 leaves at 1 (**a**–**f**) and 2 (**g**–**l**) dpi with the fungal pathogen. This figure shows representative genes with differential expression, while all expression results are provided in Appendix A. Data represent mean ± SEM; *n* = 3. Statistical analysis for each group was performed by comparing with non-inoculated (NI) controls for non-pathogen infection or NI+CC for pathogen infection. Symbols indicate statistical significance (*t*-test, * *p* < 0.05).

**Figure 4 plants-13-02495-f004:**
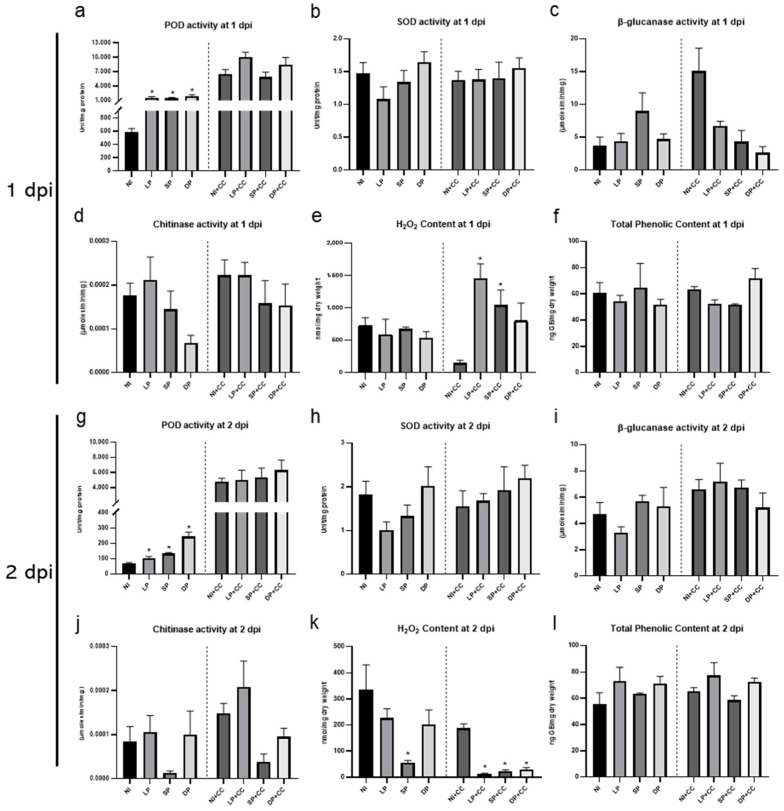
The enzyme activities and hydrogen peroxide and phenolic content of *V. radiata* CN72 leaf after 1 (**a**–**f**) and 2 (**g**–**l**) dpi. Peroxidase (**a**,**g**), superoxide dismutase (**b**,**h**), *β*-glucanase (**c**,**i**), chitinase (**d**,**j**), hydrogen peroxide (**e**,**k**), total phenolic content (**f**,**l**). Data represent mean ± SEM; *n* = 3. Statistical analysis for each group was performed by comparing with non-inoculated controls (NI) for non-pathogen infection or NI+CC for pathogen infection. Symbols indicate statistical significance (*t*-test, * *p* < 0.05).

**Figure 5 plants-13-02495-f005:**
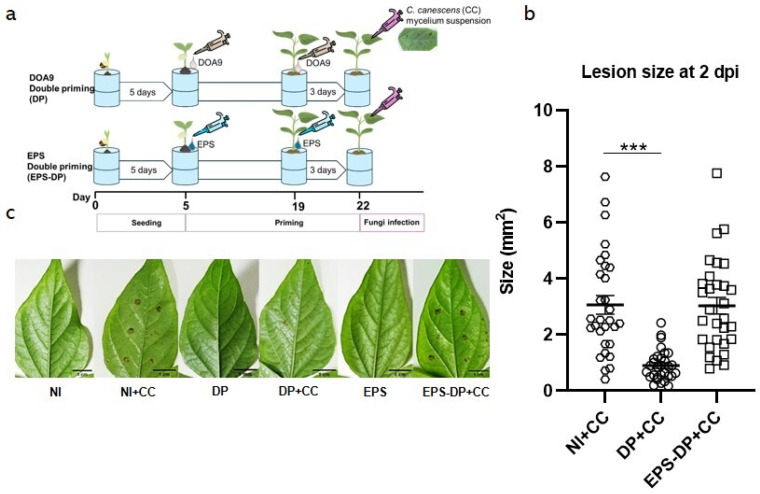
The effect of exopolysaccharide (EPS) on induced resistance of *V. radiata* CN72 against *C. canescens* (CC) infection at 2 dpi. The time schedule for seeding, DOA9, EPS priming, and CC infection is presented in (**a**). Lesion size is presented in (**b**) and phenotype of lesions on leave after infection with CC at 2 dpi is presented in (**c**). Non-DOA9-inoculated plants were used as controls (NI); treatments were non-DOA9-inoculated plants with *C. canescens* infection (NI+CC), double priming *C. canescens* infection (DP), and EPS using double priming *C. canescens* infection (DP+CC and EPS-DP+CC). Data represent mean ± SEM; *n* =  30. Symbols indicate statistical significance (*t*-test, *** *p* < 0.01).

**Figure 6 plants-13-02495-f006:**
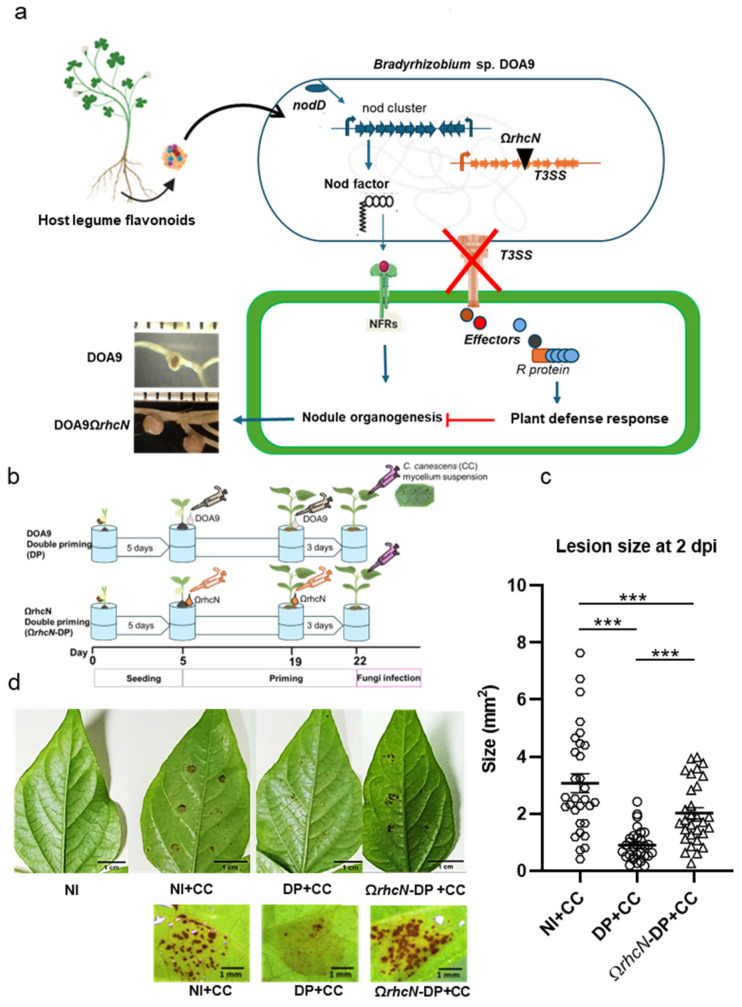
The effect of T3SS on nodule formation and induced resistance of *V. radiata* CN72 to *C. canescens* (CC) at 2 days post fungi infection. A schematic of nodulation mechanisms of *Bradyrhizobium* sp. DOA9, including nod-factor dependence and T3SS, as well as nodule phenotypes in *V. radiata* after inoculation with DOA9 wild-type and derivative mutant Ω*rhcN*, is presented in (**a**). The time schedule for seeding, DOA9, Ω*rhcN* priming, and CC infection is presented in (**b**). Lesion size is presented in (**c**) and the phenotype of lesions on leaves after infection with CC at 2 dpi is presented in (**d**). Non-DOA9-inoculated plants were used as controls (NI); treatments were non-DOA9-inoculated plants with *C. canescens* infection (NI+CC); double priming *C. canescens* infection (DP); and Ω*rhcN* using double priming *C. canescens* infection (DP+CC and Ω*rhcN*-DP+CC). Data represent mean ± SEM; *n* = 30. Symbols indicate statistical significance (*t*-test, *** *p* < 0.01).

**Figure 7 plants-13-02495-f007:**
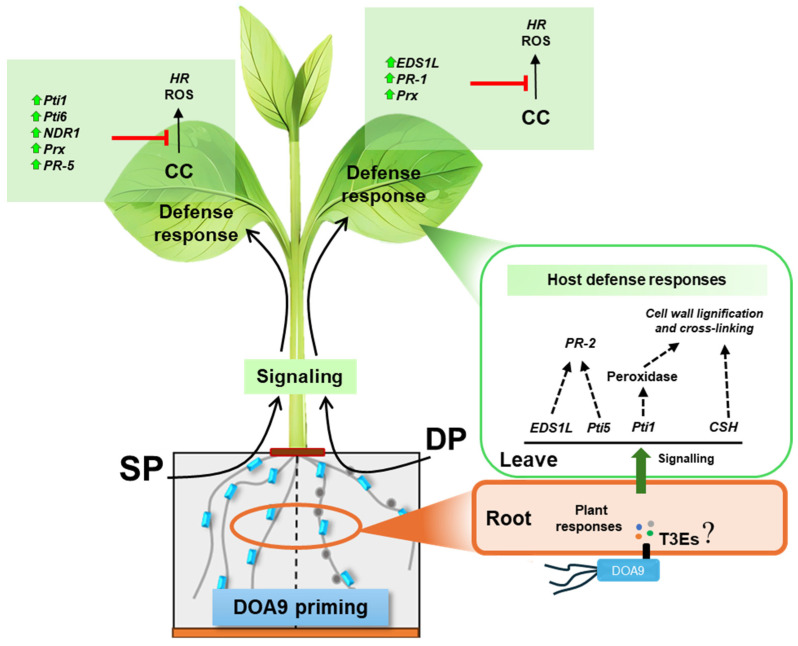
Schematic overview of the proposed mechanisms of plant defense reaction mediated by DOA9 priming on *V. radiata* CN72 roots. Solid arrows indicate known pathway, dotted arrows indicate predicted pathway depending on the expression results of this study. Green arrows beside the gene names mean upregulation and red line with blunt head means inhibition.

**Table 1 plants-13-02495-t001:** The description of bacterial strains and primers for qRT-PCR analysis in this study.

Bacterial Strains	Descriptions	Reference
*Bradyrhizobium* sp. DOA9	Wild-type strain, isolated from paddy field using *A. americana* as trap legume	[35]
Ω*rhcN*	*rhcN* mutant of DOA9 strain obtained by integration of pVO155-Sm-npt2-gfp; Sm^r^ Sp^r^ Km^r^	[35]
**Gene Names**	**Primer Names**	**Sequence 5′ to 3′**
Pti1-like tyrosine-protein kinase At3g15890 (*Pti1*)	Pti1.F PTi1.R	F: CAGCAGAAAGTGGCAGAGGA R: CGCTATAGTGTCGACCCCAC
Pathogenesis-related genes transcriptional activator Pti5 (*Pti5*)	Pti5.F Pti5.R	F: AGGCCAATGCTCTCTCCAAC R: CTGGAAAGTGCCGAGCCATA
Pathogenesis-related genes transcriptional activator Pti6-like (*Pti6*)	Pti6.F Pti6.R	F: CTGACTCCGACCACGAACAA R: TTGGGCCTTCTGCACTTTGA
Pathogenesis-related protein 1 (*PR-1*)	PR-1.F PR-1.R	F: TGAATGGACACAACCCTGCA R: GATGCCACCACCGTCTGAAT
Pathogenesis-related protein 2 (*PR-2*)	PR-2.F PR-2.R	F: GGCCCTGGAACCATCAAGAA R: TCTGCAGTGTTTGGCAAAGC
Chitinase 10 (*PR-3*)	PR-3.F PR-3.R	F: ACGACGTGATGGTTGGGAAA R: AGTGTCGACGTTGAACAGCT
Pathogenesis-related protein 4 (*PR-4*)	PR-4.F PR-4.R	F: CAGAGTTACACGGGTGGGAC R: CGTCCAAATCCAACCCTCCA
Thaumatin-like protein 1b (*PR-5*)	PR-5.F PR-5.R	F: CGATGTCCGTAACCCCACAA R: CCGTTGGTGGACATGTCTCA
Peroxidase 10 (*Prx*)	Prx.F Prx.R	F: AGGCTCTTCGACTTTGGTGG R: AAAAGAGCCTGGTCCGACTG
NDR1/HIN1-like protein 3 (*NDR1*)	NDR1.F NDR1.R	F: TGACCAAGGCCACAAGAACA R: GCGACGACACTGAACAATGG
Protein EDS1 (*EDS1*)	EDS1.F EDS1.R	F: ATAGCAGGTGTGTGGGACGA R: TCCGCGTAATGTCTCCCATG
Hypersensitive-induced response protein 2-like (*HR*)	HR.F HR.R	F: GCGGCAAGCCATAGTTGATG R: GGAAGCACCGATGTCCTTCA
Chalcone synthase 17-like (*CHS*)	CHS.F CHS.R	F: ATGAAATCCGGCAGGCTCAA R: GCACATGCGCTGGAATTTCT
Actin (*ACT*)	Actin.F Actin.R	F: CAGTGTCTGGATTGGAGGCT R: GTCCTCGACCACTTGATG

## Data Availability

Data are available on request from the first author.

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
