# Peer review of "Enhancing Resistance to Cercospora Leaf Spot in Mung Bean (Vigna radiata L.) through Bradyrhizobium sp. DOA9 Priming: Molecular Insights and Bio-Priming Potential"

_plants, 2024, doi:10.3390/plants13172495_

Round 1

Reviewer 1 Report

Comments and Suggestions for Authors

What was the difference between ΩrhcN mutants originating from the original strain? How many of them were used in the experiments? the information on the mutant strain used (lines 603-607) should be given earlier. Some information is provided in lines 316-317: a T3SS lacking- 316 mutant of DOA9 (ΩrhcN) but it should be shown in the part describing materials and methods.

page 16 - what does it mean: "necessary antibiotic supplementation"?, give a piece of detailed information;

line 633 - "Infected leaves were corrected and stained" corrected or collected? if corrected - why and how was it done?

line 612 and 640 - Half a gram/ One hundred milligrams - please use or add numbers (0,5 g/ 100 mg)

I notice several gaps in the text, e.g. lines 407, 523, 531, and others - please remove them. 

There is a symbol "DOA9-WT in Figure 6, I guess it means wild type but it is not used in any other place and is not explained.

Author Response

Reviewer 1

Comment 1: What was the difference between ΩrhcN mutants originating from the original strain?

Response 1: The ΩrhcN is a derivative mutant of DOA9 which was mutated the rhcN gene by insertion method. The rhcN gene encodes an essential ATPase required for the proper functioning of the T3SS injectisome. Thus, this mutation impacts the release of type 3 effector proteins in the DOA9 strain. In contrast, the T3SS is fully operational in the DOA9 wild-type strain. However, to clarify in this point, we explained more about this mutant strain in line 318-325, page 10. As shown below

“To investigate the role of the T3SS in plant resistance against C. canescens, we conduct-ed a double-priming experiment using a T3SS-deficient mutant of DOA9, designated as ΩrhcN. This mutant strain has a disrupted rhcN gene, which encodes a crucial ATPase necessary for the proper function of the T3SS injectisome. Unlike the wild-type DOA9, ΩrhcN is unable to secrete type III effector proteins into plant cells, leading to the formation of effective nodules (Figure 6a). The double-priming experiment was performed in comparison with the wild-type DOA9 strain (Figure 6b).”

Comment 2: How many of them were used in the experiments? the information on the mutant strain used (lines 603-607) should be given earlier. Some information is provided in lines 316-317: a T3SS lacking- 316 mutant of DOA9 (ΩrhcN) but it should be shown in the part describing materials and methods.

Response 2: The details of bacterial strains used in this study have been descripted in table 1 and line 575.

Comment 3: Page 16 - what does it mean: "necessary antibiotic supplementation"?, give a piece of detailed information.

Response 3: Done, thanks for your suggestion in this point. We designed to change “necessary antibiotic supplementation” to be “antibiotic supplementation (200 µg/ml of spectinomycin)” in this manuscript. in line 577.

Comment 4: Line 633 - "Infected leaves were corrected and stained" corrected or collected? if corrected - why and how was it done?

Response 4: Done, we changed as your suggestion, the we edited the sentence “Infected leaves were corrected and stained by collecting and boiling them for 2 min in an alcoholic lactophenol trypan blue solution, as described by [76]’’ to be “Infected leaves were collected and stained by boiling them for 2 min in an alcoholic lactophenol trypan blue solution, as described by [76]”. in line 635-636.

Comment 5: Line 612 and 640 - Half a gram/ One hundred milligrams - please use or add numbers (0,5 g/ 100 mg)

Response 5: Done, we changed the word “Half a gram/ One hundred milligrams” to be “0.5 g and 100 mg”. in line 615, 616 and 640.

Comment 6: I notice several gaps in the text, e.g. lines 407, 523, 531, and others - please remove them. 

Response 6: Done, we have removed the gaps in the text following the suggestion in lines 407, 523, 531, and others place.

Comment 7: There is a symbol "DOA9-WT in Figure 6, I guess it means wild type but it is not used in any other place and is not explained.

Response 7: Done, thanks for your suggestion in this point. We edited the word DOA9-WT to be DOA9. in figure 6.

Reviewer 2 Report

Comments and Suggestions for Authors •Authors used the Bradyrhizobium sp. strain DOA9 as treatment to increase plant defense to pathogen. The result provides a guide for the sustainable alternative to chemical fungicide applications.   •the Bradyrhizobium sp. strain DOA9 cannot  successfully form  normal symbiotic nodules, which means DOA9 can not infect V. radiata root and might induce defense response of host plant. Therefore, it is normal to activate defense-related genes when V. radiata root  was inoculated with DOA9. Please provide the phenotype of plant susceptibility or resistance to diseases after inoculation, instead of only the lesion size.   •Figure 5, please add the treatment only using EPS.
•Author mentioned that DOA9 could be used as a biocontrol agent to manage CLS in V. radiata CN72. Do all rhizobia that do not form a symbiotic nitrogen fixation system with mung beans can be used as a biocontrol agent for it? Comments on the Quality of English Language

Minor editing of English language required.

Author Response

Comment 1: Authors used the Bradyrhizobium sp. strain DOA9 as treatment to increase plant defense to pathogen. The result provides a guide for the sustainable alternative to chemical fungicide applications. The Bradyrhizobium sp. strain DOA9 cannot successfully form normal symbiotic nodules, which means DOA9 can not infect V. radiata root and might induce defense response of host plant. Therefore, it is normal to activate defense-related genes when V. radiata root was inoculated with DOA9. Please provide the phenotype of plant susceptibility or resistance to diseases after inoculation, instead of only the lesion size.

Response 1: Thank you for pointing this out. According to the nodulation phenotype of DOA9, necrotic nodules were induced by DOA9 priming, resulting to lack the ability of nitrogen fixation. Therefore, this experiment provided the plant medium with nitrogen source (2 mM KNO3) to promote the plant growth for both NI and inoculated-plants, as mentioned in material and methods page 16, line 568. The plant phenotype of the plant with DOA9 priming did not different from the non-inoculated plant (NI). Moreover, this research focus on the resistant trait. However, to clarify this point we explained more about plant phenotypes of plant with DOA9 priming compared with non-inoculated plant (NI). See in line 104-106, page 3.

“The phenotype of plants subjected to DOA9 priming did not differ from that of non-inoculated (NI) plants because DOA9 did not enhance plant growth. Thus, this experiment included a nitrogen source in the plant medium to ensure consistent growth across all treatments.

Comment 2: Figure 5, please add the treatment only using EPS.

Response 2: Thank you for your suggestion, we have added the left after priming with EPS but were not infected with CLS. in figure 5. 

Comment 3: Author mentioned that DOA9 could be used as a biocontrol agent to manage CLS in V. radiata CN72. Do all rhizobia that do not form a symbiotic nitrogen fixation system with mung beans can be used as a biocontrol agent for it?

Response 3:  Not all rhizobium strains that do not form a symbiotic nitrogen fixation relationship with mung beans can be used as biocontrol agents for them. The effectiveness of rhizobia as biocontrol agents depends on specific strains and their mechanisms, which include competition for nutrients, production of antibiotics, and secretion of enzymes that degrade pathogen cell walls. Therefore, the potential of non-symbiotic rhizobia to act as biocontrol agents needs to be evaluated on a case-by-case basis. However, in this study, DOA9 functions as a bio-priming agent rather than a biocontrol agent because DOA9 does not directly inhibit pathogens through secreted compounds. Instead, it induces defense mechanisms in the plant to prevent pathogen invasion. Consequently, we propose changing the title and replacing the term "biocontrol" with "bio-priming" throughout this manuscript.

Reviewer 3 Report

Comments and Suggestions for Authors

In this manuscript, the authors explore the potential of Bradyrhizobium sp. strain DOA9 to augment the resistance in mung bean against Cercospora leaf spot (CLS) via root priming. They found that double priming with the strain DOA9 significantly reduces the size of the lesions on mung bean infected leaves by activating or amplifying defense-related genes, in addition to suppressing the inhibition of PR-5, and that T3 secretion system (T3SS) of DOA9 may be partially implicated in the resistance of V. radiata CN72. They concluded that the strain DOA9 primes the defense mechanisms of V. radiata CN72, providing a sustainable alternative to chemical fungicides for managing CLS.

The manuscript is clear, and the results support the conclusions.

A minor concern was found in the abstract: POD should be defined in line 22.

Author Response

Response to Reviewer 3

Manuscript plants-3100467

In this manuscript, the authors explore the potential of Bradyrhizobium sp. strain DOA9 to augment the resistance in mung bean against Cercospora leaf spot (CLS) via root priming. They found that double priming with the strain DOA9 significantly reduces the size of the lesions on mung bean infected leaves by activating or amplifying defense-related genes, in addition to suppressing the inhibition of PR-5, and that T3 secretion system (T3SS) of DOA9 may be partially implicated in the resistance of V. radiata CN72. They concluded that the strain DOA9 primes the defense mechanisms of V. radiata CN72, providing a sustainable alternative to chemical fungicides for managing CLS. The manuscript is clear, and the results support the conclusions.

Comment 1: A minor concern was found in the abstract: POD should be defined in line 22.

Response 1: Thank you for your suggestion on this point, peroxidase (POD) activity has been defined in the abstract, Line 22.
